# Graphene and Polyethylene: A Strong Combination Towards Multifunctional Nanocomposites

**DOI:** 10.3390/polym12092094

**Published:** 2020-09-15

**Authors:** Mar López-González, Araceli Flores, Fabrizio Marra, Gary Ellis, Marián Gómez-Fatou, Horacio J. Salavagione

**Affiliations:** 1Departamento de Química Física de Polímeros, Instituto de Ciencia y Tecnología de Polímeros, ICTP-CSIC, c/Juan de la Cierva 3, 28006 Madrid, Spain; mar@ictp.csic.es; 2Departamento de Física, Elastómeros y Aplicaciones Energéticas, Instituto de Ciencia y Tecnología de Polímeros, ICTP-CSIC, c/Juan de la Cierva 3, 28006 Madrid, Spain; araceli.flores@csic.es (A.F.); gary@ictp.csic.es (G.E.); magomez@ictp.csic.es (M.G.-F.); 3Department of Astronautical, Electrical and Energy Engineering, Sapienza University of Rome, Via Eudossiana 18, 00184 Rome, Italy; fabrizio.marra@uniroma1.it; 4Research Center for Nanotechnology Applied to Engineering of Sapienza (CNIS), SNNLab, Sapienza University of Rome, Piazzale Aldo Moro 5, 00185 Rome, Italy

**Keywords:** polyolefin, carbon nanofillers, electrical conductivity, gas barrier, mechanical properties, thermal stability

## Abstract

The key to the preparation of polymer nanocomposites with new or improved properties resides in the homogeneous dispersion of the filler and in the efficient load transfer between components through strong filler/polymer interfacial interactions. This paper reports on the preparation of a series of nanocomposites of graphene and a polyolefin using different experimental approaches, with the final goal of obtaining multifunctional materials. A high-density polyethylene (HDPE) is employed as the matrix, while unmodified and chemically modified graphene fillers are used. By selecting the correct combination as well as the adequate preparation process, the nanocomposites display optimized thermal and mechanical properties, while also conferring good gas barrier properties and significant levels of electrical conductivity.

## 1. Introduction

At the frontiers of materials technology, the concept of synergistically combining various materials of very different nature is one of the most successful approaches to achieve multifunctionality. In this respect, the appropriate combination of graphene with polyolefins, two families of materials with very different but complementary properties, provides a route to the creation of versatile materials with countless applications, because it not only refers to the combination of two compounds but in the assembly of two families of materials.

On the one hand, polyolefins are the most widely exploited classes of polymers on the planet due to their good chemical and physical properties, together with their low production costs, easy processing, extraordinary versatility, and their recyclability potential and application as sustainable materials. The annual global production of polyolefins in 2015 was estimated at over 180 million tons [1] and with a compound annual growth rate (CAGR) of nearly 6% is expected to reach over 250 Mt by 2025. This family of polymers is principally applied in food packaging, automotive components, cosmetics, medical products, household chemicals, to name but a few.

On the other hand, graphene should not be considered as a single material as it encompasses a vast range of materials due to its broad range of unique and extraordinary properties, including flexibility with superior mechanical properties (Young’s modulus ~1 TPa) [2], high thermal conductivity (~5000 Wm^−1^K^−1^) [3], high electrical conductivity (>6 × 10^5^ Scm^−1^) [4], high specific surface area (estimated > 2600 m^2^g^−1^) [5], optical transparency [6] along with several non-conventional electronic properties including the anomalous quantum Hall effect and massless Dirac fermions together with ballistic electronic transport [7].

In this respect, the effective combination of graphene with polyolefins constitutes a powerful route for the creation of new materials with superior properties [8] and the key to success resides in the homogeneous dispersion of graphene and in the efficient load transfer between components through strong filler/polymer interfacial interactions [9].

Similar to other polymer matrices, the combination of graphene with polyolefins generally leads to mechanically stronger nanocomposites with improved thermal stability [10,11,12,13,14,15,16]. The former is attributed to a barrier effect of the nanomaterial that effectively hinders the transport of volatile decomposition products from the bulk of the polymer to the gas phase, hence retarding the degradation process [13]. Mechanical enhancements are primarily associated with the inherent properties of the nanomaterial with a high Young’s modulus (~1 TPa for defect-free graphene) and intrinsic strength (130 GPa) far superior to any polymer [17]. However, of particular interest are other properties, such as the electrical conductivity, gas barrier, and electromagnetic interference (EMI) shielding, for which the distribution of graphene and the strength of the interface play a much more decisive role.

Electrical conductivity represents the biggest challenge for this type of materials since conductive nanocomposites find applications in many areas including field effect transistors, solar cells, energy storage devices, anti-static coatings, and electromagnetic interference (EMI) shielding [18]. However, there are only a few studies that have delivered polyolefin-based nanocomposites with reasonable electrical conductivity. Early work on such materials, described the preparation of nanocomposites of high-density polyethylene (HDPE) with graphene covalently modified ad hoc [19,20]. We developed a series of chemical strategies to provide graphene with pendant short polyethylene brushes to improve its dispersion in the HDPE matrix. This chemical functionalization approach along with the engineering of the HDPE/graphene interface during the preparation process were crucial to obtain nanocomposites with improved mechanical properties and thermal stability as well as with good levels of electrical conductivity [19,20]. A different chemical approach based on the covalent modification of graphene oxide with vinylsilane groups and in situ polymerization of ethylene for the preparation of electrically conductive graphene/polyethylene nanocomposites has been recently reported [21]. Additionally, graphene oxide has been used as a platform for chemical functionalization to improve the dispersion of graphene into polyethylene. Macoscko et al. have capitalized on the chemistry of the hydroxyl and epoxide groups on graphene oxide (GO) to graft alkyl chains and benzyl moieties to GO. After mixing with different polyethylene matrices and thermal reduction they achieve good electrical conductivities for the nanocomposites [22]. Beyond covalent functionalization, some noncovalent compatibilization approaches have also succeeded in the preparation of electrical conductive graphene/polyethylene nanocomposites [23,24]. In addition, some studies have reported good electrical conductivity values and low percolation threshold on nanocomposites with ultra-high molecular weight polyethylene (UHMWPE) [25,26,27].

Beyond electrical conductivity, gas barrier is also one of the most explored functionalities for this kind of materials [28] due to the important role that polyethylene plays in packaging for food and pharma applications. In order to solve the problem of high permeability to gases, the incorporation of laminar nanoparticles such as graphene has been intensely investigated [10,11,12,14,15,16,29,30,31,32]. Graphene nanosheets are intrinsically impermeable to almost all gas molecules [33] and also present high aspect ratio structures, which make them ideal to engineer tortuous pathways to decrease the molecular diffusion rates through polymeric materials [34,35]. Therefore, the efficient, aggregate-free dispersion of graphene is fundamental to achieve maximum tortuosity for gas molecules. It has been recently reported that the functionalization of graphene oxide (GO) with non-polar alkyl chains and the appropriate selection of the mixing procedure promotes strong interfacial adhesion with a UHMWPE matrix, which leads to a positive effect on gas barrier properties [31]. The use of very defective and polar GO with hydrophobic matrices normally leads to a modest or null effect on the barrier properties of the nanocomposites [30]. For graphene nanoplatelets melt-compounded with low density polyethylene (LDPE), a decrease in the permeability of carbon dioxide of up to 65% was observed [16]. Although the increase in tortuosity by the presence of 2D laminates is widely accepted, it has been suggested that other factors related to characteristics of the polymer also contribute to the reduction of permeability [32]. Checchetto et al. have reported a reduction of the permeability to hydrogen, nitrogen, and carbon dioxide by a factor of two for nanocomposites of LDPE and graphite nanoplatelets. They compared the experimental results with those predicted by phenomenological models and concluded that rigidity of layers around the filler particles plays a relevant role [32].

The objective of the present work is the evaluation of different approaches for the preparation of graphene/polyethylene nanocomposites in order to select the best to obtain multifunctional materials. Experimental variables such as graphene type, mixing approach, and other physical parameters are explored in order to achieve the optimal conditions for specific functionalities. Gas barrier and electrical properties are analysed in detail and discussed together with thermal and mechanical properties as a function of the type of graphene filler (neat or functionalized) and of the mixing approach used to prepare the nanocomposites.

## 2. Experimental

### 2.1. Materials

Graphene nanoplatelets (GNPs) with reference N002-PDR were purchased from Angstron Materials (Dayton, OH, USA). According to the specifications of the manufacturer, these GNP present fewer than three graphene layers with a maximum lateral size of 10 µm. The carbon content was ≥95.0%, with an oxygen content ≤2.50%.

A commercial high density polyethylene (HDPE) was employed as the polymer matrix, with a mass flow rate (MFR) of 1.5 × 10^−5^ m^3^/10 min (2.16 kg; 190 °C), and a density of 945 kg·m^−3^, kindly provided by Repsol S.A. (Móstoles, Spain).

Oxygen and hydrogen (99.999%), nitrogen (99.9999%), carbon dioxide (99.998%), and methane (99.9995%) gases from Praxair (Madrid, Spain) were used in the permeation experiments.

### 2.2. Modification of Graphene

The graphene nanoplatelets were click-functionalized with short polyethylene brushes according to a previously reported procedure [20]. The modified-graphene product was denominated GPE. The actual composition of GPE was estimated by thermogravimetric analysis (TGA), where the mass loss due to the elimination of the PE brushes corresponds to 18%. Hence, GPE contains 82 wt % of graphene.

### 2.3. Preparation of Nanocomposites

Nanocomposites were prepared by two different processing methods. In the first, modified graphene (GPE) and neat graphene (G) were mixed with HDPE in hot xylene (90 °C) under vigorous stirring and precipitated in methanol, filtered, washed, and thoroughly dried under vacuum. Nanocomposites prepared from solution with a final GPE or G loading of ~1 wt % were denominated GPE1S and G1S, respectively. In the second method, nanocomposites were prepared by a melt-blending process, performed in a Haake Minilab extruder (Thermo Scientific, Waltham, MA, USA) operating at 180 °C, with a rotor speed of 100 rpm, using a mixing time of 10 min. Nanocomposites prepared from melt compounding with ~1 wt % and 3 wt % GPE loading were denominated GPE1M and GPE3M, respectively. Nanocomposites with around 1 wt % of neat graphene were also prepared by melt-blending for comparison purposes, and denominated G1M. Table 1 includes the graphene content in wt % for all nanocomposites, determined by TGA. Finally, films with a thickness in the range 80–250 μm of all nanocomposites and neat HDPE (140 µm) were obtained by compression molding at 170 °C and a pressure of 150 bar.

### 2.4. Characterization

The dispersion of the fillers in the HDPE matrix was examined by scanning electron microscopy (SEM) using a Hitachi SU8000 field emission microscope (Tokyo, Japan). The nanocomposite samples were cryofractured from the corresponding films and images were collected at 0.8 kV, using a secondary electron and backscattered electron detector combination.

The morphology of the nanocomposites was also examined by transmission electron microscopy (TEM). Images were obtained with a Philips Tecnai 20 microscope (Philips Electron Optics, Holland, Netherlands). Ultrathin sections of 50–100 nm in thickness were cryogenically microtomed with a diamond knife at ~−60 °C and supported on copper TEM grids. These sections were observed without further preparation to evaluate the graphene dispersion in the matrix under different processing conditions. 

Differential scanning calorimetry (DSC) measurements were carried out in a Perkin-Elmer DSC7-7700 calorimeter (Perkin-Elmer España S.L., Madrid, Spain), calibrated with indium (*T_m_* = 156.6 °C, Δ*H_m_* = 28.45 kJ kg^−1^) and zinc (*T*_m_ = 419.47 °C, Δ*H*_m_ = 108.37 kJ kg^−1^). Samples, of approximately 10 mg weight were tested in aluminium pans under an inert nitrogen atmosphere flow at a rate of 2.5 × 10^−5^ m^3^ min^−1^. For all samples, heating scans were recorded from 40 °C to 160 °C at a rate of 10 °C min^−1^ and the melting temperature, *T*_m_, was determined as the maximum of the melting endotherm observed during the scan. The degree of crystallinity, *X*_c_, was obtained by dividing the crystallization enthalpy of the nanocomposites, obtained from the DSC curves (corrected for the amount of HDPE), by the value for 100% crystalline HDPE taken to be 286.2 Jg^−1^ [36].

TGA was carried out using a TA Instruments Q50 thermobalance (Waters Cromatografía, S.A., Cerdanyola del Vallès, Spain) between 50−800 °C at a heating rate of 10 °C min^−1^, under an inert atmosphere (nitrogen, 60 cm^3^ min^−1^). Samples were analysed using TA Instruments Universal Analysis 2000 software (version 4.5A, Build 4.5.0.5).

Mechanical properties were evaluated using instrumented indentation testing. Portions of the films (typically 3 × 2 mm^2^) were glued onto a metallic cylindrical holder that was placed on the sample stage of a G200 nanoindenter (KLA Tencor, Milpitas, CA, USA). During the loading cycle, the load *P* was incremented at a constant *Ṗ*/*P* ratio (where *Ṗ* = *dP/dt*) to ensure a constant indentation strain rate (selected to be 0.05 s^−1^). At the same time a small oscillating force of 75 Hz was superimposed to the quasi-static loading and produced an oscillation displacement of 1 nm. This allowed a continuous measure of the contact stiffness, *S*, on the basis of the phase lag between the oscillating force and the indenter penetration and adopting a simple harmonic oscillator model to describe the dynamic response of the instrument and sample [37]. Finally, storage modulus and hardness were determined assuming elastic-viscoelastic correspondence [37,38]
(1)$$\frac{E´}{1-\nu^{2}}=\frac{\sqrt{\pi }}{2}\frac{1}{\beta \sqrt{A_{c}}}S$$
(2)$$H=\;\frac{P}{A_{c}}$$

Poisson’s ratio, *ν*, is taken to be 0.35 in all cases, *β* = 1.034 for a Berkovich indenter and *A_c_* is the contact area at *P*. The area function describing *A_c_* was calibrated as a function of the contact penetration depth, *h_c_*, using a fused silica standard and *h_c_* was estimated as described in reference [38].

DC-conductivity measurements were carried out on films dried under vacuum for 24 h. The measurements were made using a four probe setup comprising a dc low-current source (LCS-02) and a digital microvoltmeter (DMV-001) from Scientific Equipment & Services Pvt Ltd. (Roorkee, India). The temperature of the samples was regulated with an accuracy of ±0.5 °C using a PID controlled oven from Scientific Equipment & Services Inc.

The conductivity *σ* was calculated as the inverse of the resistivity, *ρ* by
(3)σ=1/ρ
where
(4)$$\rho =4.5324\;t\;\left(\frac{V}{I}\right)f_{1}f_{2}$$
with *t* being the thickness of the sample, *I* the applied current, *V* the measured voltage, *f*_1_ the finite thickness correction for thick samples on an insulating bottom boundary, and *f*_2_ the finite width correction.

Pure gas permeation experiments of oxygen, nitrogen, hydrogen, carbon dioxide, and methane gases though all the membranes were carried out at a pressure of 1 bar and 30 °C using a thermostatted experimental device described in detail elsewhere [39]. Briefly, the permeation cell is made up of two chambers separated by the membrane. A Gometrics (Barceolna, Spain) pressure transducer with a 0–10 bar range measures the pressure of gas in the high pressure, or upstream, chamber whereas the evolution of the pressure with time in the low-pressure, or downstream, chamber was monitored via a desktop computer with a MKS Baratron type 628 pressure sensor (MKS Instruments Inc., Andover, MA, USA) operating in the range of 10^−4^ to 1 mmHg. The permeation area diameter of the membranes was 2.1 cm. Before performing each experiment, the air inlet into the evacuated downstream chamber was monitored as a function of time and then subtracted from the curves representing the pressure of the permeant against time in the downstream chamber.

## 3. Results and Discussion

### 3.1. Graphene Dispersion

One of the most critical aspects that determine the performance of polymer/graphene nanocomposites is the dispersion of the nanomaterial in the matrix. The processing method has a substantial effect on this factor as does the modification of the nanofiller. In order to evaluate the morphology of the nanocomposites, all materials were examined by different microscopies. SEM was employed to determine the dispersion of graphene in the polyethylene matrix, while the degree of exfoliation of the nanofiller was observed by TEM. Figure 1 shows the SEM images of the cryogenically fractured surfaces of all samples.

A more homogeneous graphene distribution was observed when the nanocomposites were prepared from solution as opposed to those prepared from the melt, as can be seen by comparing, for example, Figure 1A,B. The influence of the covalent attachment of polymeric brushes on the graphene dispersion is clearly observed when comparing Figure 1A,C. The nanocomposite G1M with 1 wt % of pristine graphene (Figure 1A) prepared from the melt displayed a coarse morphology with the formation of islands of partially wrinkled graphene laminates distributed throughout the matrix. However, the nanocomposite GPE1M (Figure 1C) prepared under the same processing conditions and with the same filler content where the graphene was modified with polyethylene (PE) brushes showed a more homogeneous distribution of smaller sized islands. Larger aggregates were formed when the graphene content increased and some degree of orientation seemed to develop due to the shear and elongational forces during the melt mixing process (Figure 1D).

In order to further investigate the nanocomposites morphology, TEM images of all materials were obtained. As shown in Figure 2, graphene sheets appear to be partially exfoliated with some wrinkled or folded laminates, but generally well dispersed within the polymer matrix in all nanocomposites. Only in the nanocomposite processed by extrusion with the highest filler content were larger aggregates observed with some degree of orientation (Figure 2D).

The influence of processing conditions and particle size, content and functionalization on the dispersion of graphene have been previously reported for HDPE nanocomposites as determined by SEM and TEM studies [11,15,19,20]. It can be clearly shown that our results corroborate the importance of these parameters in the level of graphene dispersion in HDPE. 

### 3.2. Thermal Behavior

The thermal stability of all nanocomposites was investigated by TGA under a nitrogen atmosphere. The thermogravimetric (TG) results are shown in Figure 3. Table 1 presents the characteristic degradation temperatures for all samples. An enhancement in thermal stability was observed for all nanocomposites, independent of the processing conditions and the type of graphene, with an increase in all characteristic degradation temperatures compared with HDPE. With increasing filler content, the initial degradation temperature (*T_i_*) and the temperatures corresponding to 10% weight loss (*T*_10_) and to the maximum rate of weight loss (*T_max_*) varied from 406 °C, 450 °C, and 485 °C, respectively, for the neat matrix to maximum values of 444 °C, 466 °C, and 490 °C for the highest graphene concentration (GPE3M, 3 wt % GPE). These results demonstrate that well-dispersed graphene can effectively hinder the diffusion of the degradation products slowing down the decomposition process of the nanocomposites. The barrier effect of graphene in these materials will be discussed in detail below in the permeability section. Regarding the processing conditions, a clear influence was found with higher degradation temperatures for those nanocomposites prepared from solution (GPE1S compared to GPE1M) due to a more homogeneous dispersion of the carbon filler in the matrix, as was demonstrated by SEM. When comparing the incorporation of modified graphene with the non-modified filler in the nanocomposites prepared from solution, an increase in the characteristic temperatures was observed for those incorporating GPE as a result of the enhanced interactions between the polymer and the filler that favour a more homogeneous dispersion of the carbon nanomaterial. 

An enhancement in thermal stability with the incorporation of graphene has been previously found for polyolefin nanocomposites [8,13] and, in particular, for HDPE nanocomposites [11,19]. It can be noted that the increase observed in *T_i_* for GPE3M is one of the highest observed when compared to those described in the literature for HDPE/graphene nanocomposites.

The intrinsic nanostructure of a semicrystalline polymer matrix can have a significant influence on some of the properties of its nanocomposites, especially on the mechanical performance and gas permeability of the membranes. Therefore, the influence of graphene on the crystallization processes of HDPE was studied by DSC and their thermal parameters were determined. Table 1 includes the degree of crystallinity (*X_c_*) and melting temperature (*T_m_*) for all the materials investigated. It can be noticed that the incorporation of graphene does not significantly alter either parameter in the nanocomposites, regardless the nature of the filler or the preparation conditions.

Nevertheless, a very small increase in *X_c_* and *T_m_* can be observed in GPE1M and GPE1S compared with the nanocomposites with unfunctionalized graphene. As the amount of GPE increases from 1 wt % to 3 wt % in the nanocomposites prepared from the melt, a slight decrease in both parameters is observed. It has been previously demonstrated that graphene can exert a nucleating effect on the crystallization of a polyolefin matrix, especially in isotactic polypropylene [8,13]. A nucleating effect of the nanofiller as well as the retardation of the crystallization process have been reported for HDPE/graphene nanocomposites depending on concentration, dispersion and functionalization of the nanofiller [11,15,19,20,21]. For example, Honaker et al. reported a significant increase in the matrix crystallinity in HDPE/graphene nanocomposites prepared by melt mixing at low concentrations of the filler (up to 2 wt %) due to a nucleating effect [15]. As the graphene content increased crystallinity started to decrease due to the reduction in the mobility and diffusion of the HDPE chains and the presence of filler agglomerates [15]. Our results show a similar trend. However, the changes in crystallinity are very small and it is expected that they would not have a large effect on the mechanical and barrier properties.

### 3.3. Mechanical Properties

The mechanical performance of the nanocomposites was investigated by indentation testing. Figure 4 illustrates, as an example, the storage modulus, *E′*, and hardness, *H*, behavior as a function of indenter displacement into the surface of HDPE and two of the nanocomposites. A large data scatter is found at small *h* values below ≈500 nm most probably due to surface roughness. This effect is minimized as the indenter progresses towards the bulk and one can see fairly constant *E*′ and *H* values beyond *h* ≈ 500 nm. Such behavior indicates that no significant micrometer-scale inhomogeneities associated to changes in the matrix structure or in the filler distribution appear across the sample thickness. Moreover, the error bars in Figure 4 represent the standard deviation of the mechanical data retrieved at different locations along the surface (≈40 indentations). It can be seen that values of the standard deviation are quite small at large indentation depths: for each *h* value, the standard deviation is around 2–6% of the mean *E′* and *H* values. It is also noteworthy that the incorporation of graphene has no influence on the data dispersion. Hence, the results suggest that the dispersion of the filler is also homogeneous at a micrometer scale along the surface. To make this point consistent with the dispersion of graphene platelets shown in the SEM micrographs of Figure 1, one should recall that the volume of deformation is not limited to the region immediately next to the indenter-sample contact but encompasses a large volume that typically extends up to ≈10 times the indentation depth for the plastic field and in the range of ≈20 times the indenter displacement for the elastic limit [40,41].

Figure 4 also shows that the *E′* and *H* values increase with the incorporation of graphene. In order to conveniently analyse the influence of graphene on the mechanical behavior, Figure 5 shows the indentation *E′* and *H* values at *h* = 2 μm for all the nanocomposites studied (see also Table 2). Data include different types of filler (graphene with or without low molecular weight PE brushes) and preparation processes (solution mixing or melt extrusion). In the first place, a small mechanical increase is found for all graphene loadings around 1 wt %. A closer inspection reveals that the incorporation of neat graphene is most efficient from a mechanical point of view when a solution step is included in the preparation process of the nanocomposite (GS sample), as compared to melt extrusion (GM). Taking into account that the GM and GS samples exhibit quite similar values for the degree of crystallinity (see Table 1), the differences in mechanical behavior could be associated with a better dispersion of graphene at the nanometer scale for the solution case, as indicated earlier on the basis of SEM studies (Figure 1). An analogous analysis can be carried out for the two nanocomposites prepared under melt extrusion (GM and GPEM) and showing similar crystallinity values (*X_c_* = 55% and 60%, respectively; see Table 1). In this case, comparison of the *E′* and *H* values points towards slightly higher values for the case of modified graphene which, according to SEM analysis (Figure 1), could be related to improved filler dispersion arising from the interaction of the PE brushes attached to the graphene surface with the chains of the PE matrix. Hence, interestingly, the modification of graphene appears as a most convenient strategy to obtain mechanical enhancements similar to those attained through the solution route but via melt extrusion, which is the most attractive for industrial applications. Finally, Figure 5 includes the *E′* and *H* values for one GPEM nanocomposite with a high loading of modified graphene (3 wt %, excluding the polymer brushes). In this case, there is a significant mechanical reinforcement of HDPE by 20%, similar to that reported for other polyolefin/modified-graphene nanocomposites prepared using a solution mixing step [13]. This value is in agreement with preceding studies in other HDPE/graphene nanocomposites that typically show modulus enhancements of 15–100% for 1–3 wt % graphene loadings [8,11,15,22,29]. The merit of the present melt extruded materials is the simplicity of the preparation process that can be easily scaled up for production purposes.

### 3.4. Gas Permeability 

It is well known that gas transport through a dense membrane can be described by a diffusion-solution mechanism that involves three steps: the sorption of the gas in the membrane, which is conditioned by thermodynamic interactions between the gas and the membrane, the diffusion of the gas across the film, that describes the kinetic aspects of the transport, and the desorption of the gas at the other side of the film. Hence, the permeability of a gas through a film can be generally expressed as a product of the diffusion and the solubility of the gas in the membrane.

The curves showing the evolution of the gas transport as a function of time exhibit a transient zone at short time followed by a region at long times in which the pressure is a linear function of time. An illustrative plot of the variation of the pressure *p*(*t*) of oxygen, nitrogen, and carbon dioxide in the downstream chamber as a function of time in the GPE1M membrane is represented in Figure 6.

The evolution of the pressure of the permeant in cm of Hg is described by the integration of Fick’s second law using appropriate boundary conditions [42],
(5)$$p\left(t\right)=0.2786\;\frac{p_{0}ALST}{V}\;\left(\frac{Dt}{L^{2}}-\frac{1}{6}-\frac{2}{\pi^{2}}\overset{\infty }{\underset{n=1}{\sum }}\frac{{\left(-1\right)}^{n}}{n^{2}}\;\exp\left(-\frac{Dn^{2}\pi^{2}t}{L^{2}}\right)\right)$$
where *p*(*t*) and *p*_0_ denote the pressures of gas in the downstream and upstream chambers, respectively; *A* and *L* represent the area and thickness of the membrane in cm^2^ and cm, respectively; *T* is the absolute temperature and *V* is the volume of the downstream chamber in cm^3^, whereas *S* (in cm^3^ (STP) cm^−3^ cmHg^−1^) and *D* (in cm^2^ s^−1^) are, respectively, the solubility and diffusion coefficients.

When steady-state conditions are reached (*t*→∞), Equation (5) can be simplified and a linear relationship is found between *p*(*t*) and t described by Equation (6)
(6)$$p\left(t\right)=0.2786\;\frac{p_{0}ALST}{V}\;\left(\frac{Dt}{L^{2}}-\frac{1}{6}\right)$$

The straight line given by Equation (6) intercepts the time axis at *θ*
*= L*^2^*/6D*, where *θ* is the time-lag. Therefore, the diffusion coefficient can be determined directly from *θ* by using the following expression, as early suggested by Barrer [43]
(7)$$D=\;\frac{L^{2}}{6\theta }$$

Taking into account that the permeability coefficient is defined as *D* × *S*, this parameter can be obtained from Equation (6) by means of the expression:(8)$$\Phi =3.59\;\frac{VL}{p_{0}AT}\;\underset{t\to \infty }{\lim}\frac{dp\left(t\right)}{dt}$$
where Φ is given in Barrer [1 Barrer = 10^−10^ cm^3^ (STP) cm/(cm^2^ s cmHg)]. The apparent solubility coefficient can be calculated by
(9)$$S=\;\frac{\Phi }{D}$$

Values of the permeability, diffusion and apparent solubility coefficients for the gases in the nanocomposite membranes prepared by solution and melt mixing at 30 °C and 1 bar of pressure are shown in Table 3 and Table 4, respectively. In Table 3 and for comparative purposes, the values of these coefficients for the pure HDPE are also shown.

An inspection of these tables shows that all membranes exhibit the same behavior when comparing results for different gases. Thus, the permeability coefficients follow the trends Φ(CO_2_) > Φ(H_2_) > Φ(O_2_) > Φ(CH_4_) > Φ(N_2_). Regarding the diffusion coefficients: D(H_2_) > D(O_2_) > D(N_2_) > D(CO_2_) > D(CH_4_), closely related to the Lennard–Jones diameter of the diffusive species in such a way that the lower the diameter, the higher the diffusion coefficient. Finally, the solubility: S(CO_2_) > S(CH_4_) > S(O_2_) > S(N_2_) > S(H_2_). Evidently, carbon dioxide is the most condensable gas and exhibits the largest apparent solubility coefficient. The high value of the solubility coefficient of CO_2_ in comparison with that of H_2_ is responsible for the relatively high permeability coefficient of that gas in all nanocomposite membranes.

The errors involved in the determination of permeability and diffusion coefficients range between 3–7% and 10–17%, respectively, depending on the gas and the membrane tested.

The values of Φ and D obtained from the *p* vs. *t* curve were very similar to those estimated by fitting Equation (5) to the experimental results, as can be seen in Figure 6, where the experimental and calculated values of the evolution of pressure in the downstream chamber are shown as a function of time. As an example, the values obtained for permeability and diffusion coefficients of oxygen in GPE1M membrane, directly calculated from Equation (8) and time-lag method were 0.82 Barrer and 1.68 × 10^−7^ cm^2^/s, respectively. These values fit very well with those of 0.81 Barrer and 1.76 × 10^−7^ cm^2^/s obtained by fitting Equation (5) to the experimental results. 

As can be seen in Table 3 and Table 4, there are significant differences in all parameters depending on the type of membranes as well as on the experimental procedure employed to prepare them. Figure 7 represents the variation of the permeability (Figure 7A) and diffusion coefficients (Figure 7B) of the nanocomposites membrane with respect to the HDPE membrane. From Figure 7, it is possible to assure that the nanocomposites with functionalized graphene (GPE) present better barrier properties than those with non-functionalized graphene. In addition, in the GPEM series the higher the loading of GPE the higher the barrier effect; a permeability reduction between 40–50% for all evaluated gasses was measured for GPE3M. The differences observed between the two families of nanocomposites cannot be attributed to variations in crystallinity between the samples, which are very similar (see Table 1) but to an enhancement in the dispersion and filler/matrix interactions in the GPE series. In addition, the filler aspect ratio and its orientation within the polymer matrix relative to the diffusion direction are other factors that must be considered.

The morphology of the interface plays an important role in the overall transport properties of composite materials, and according Moore and Koros [44], five ‘cases’ to explain the relationship between morphology and transport properties could be expected. These cases are shown as different color zones in Figure 8. The Maxwell model, which assumes a perfect contact between the filler and the surrounding matrix, is represented as green zone in Figure 8 and corresponds to case 0. Case I, represented as blue zone in Figure 8, corresponds to a rigidified polymer region around the filler with a reduced permeability and a slight increase in nanocomposite selectivity in relation with pristine polymer matrix, taking into account that the ideal selectivity of a gas A with respect to other gas B can be expressed as $\alpha \;\left(\frac{A}{B}\right)=\;\frac{\Phi \;\left(A\right)}{\Phi \;\left(B\right)}$ where Φ (A) > Φ (B). Cases II and III are shown as yellow and brown zones in Figure 8, respectively, and these are characterized by having voids at the interface, although case III is a special case of case II, where the effective void thickness is on the order of the size of gas molecules. In such cases, the nanocomposites have a similar or slightly lower selectivity than neat polymer with an increase in gas permeability. Finally, cases IV and V are represented as grey and pink zones in Figure 8, respectively. In these cases, sorption of gas molecules takes place in the nanocomposite materials preventing (case IV or clogged sieves) or slowing down (case V or reduced permeability region within sieve surface) gas permeation. 

The experimental results shown in Table 3 and Table 4 for all nanocomposites described in this work are also represented in Figure 8. As can be seen, the membranes prepared from solution, independently of the nature of filler used (G1S and GPE1S) show a characteristic behavior of the formation of a rigidified polymer layer around the filler. However, the nanocomposites membranes prepared with GPE by melt-blending seem to present a typical morphology in which the gas passes through the polymer/functionalized graphene interface with reduced permeability. This fact could be explained assuming that this interface is stronger as a consequence of the better compatibility between both phases. On the contrary, the membrane prepared with non-functionalized graphene (G1M) seem to present a typical case III morphology in which the slightly lower compatibility between non-functionalized filler and HDPE would provoke some voids in the interphase region through which gas molecules can diffuse.

In order to analyze the differences in performance between GPE and G nanocomposites with the same filler loading experimental data were compared with those calculated from several theoretical models. The simplest model to predict the permeability of nanocomposite membranes is the Maxwell model that assumes an ideal filler/matrix contact where the polymer fully embedded the filler nanoparticles, but it does not feel the presence on the filler, maintaining its bulk properties [32,45,46]. According to the Maxwell model, the permeability of a nanocomposite (Φ*_NC_*) can be described by the equation
(10)$$\Phi_{NC}=\Phi_{p}\frac{\Phi_{f}+2\Phi_{p}-2\varphi_{f}\left(\Phi_{p}-\Phi_{f}\right)}{\Phi_{f}+2\Phi_{p}+\varphi_{f}\left(\Phi_{p}-\Phi_{f}\right)}$$
where Φ*_p_* and Φ*_f_* are the gas permeability of the pure polymer and the filler, respectively and *φ**_f_* is the volume fraction of the filler, which takes values of 0.023 and 0.067 for samples with 1 wt % and 3 wt % of graphene, respectively.

If we assume that the graphene sheets are not impermeable, Φ*_f_* = 0, and apply the Maxwell model, we see that the decrease of permeability is much lower than almost all the experimental data (Figure 9), whatever the diffusing molecule, confirming that factors other than the increase in the tortuous pathway of the permeating gases have to be considered. This additional factor is related to the polymer/filler interface region, where the gas permeability is markedly different from that in the bulk matrix and depends on the length of this interface [44,45]. This non-ideal polymer/filler contact induces a rigidification of the polymer chains around the filler (the interface), where the permeability is reduced with respect to that in the bulk matrix. In some cases, the formation of interface voids could occur as a result of the de-wetting of polymer chains from the filler surface in which case an increase in the permeability of the nanocomposite is observed in comparison with that of the neat polymer, as in the G1M membrane. In the case that rigidification occurs, we would expect the extent of this effect to be dependent on the chemical affinity between the filler and the matrix. The reduction in permeability in the interface region (Φ*_int_*) would be related to that in the bulk by means of an immobilization factor, β according to the equation
(11)$$\Phi_{int}=\Phi_{p}/\beta$$

Therefore, the effective permeability of the pseudo-dispersed phase (rigidified region plus dispersed particles) can be expressed by the equation
(12)$$\Phi_{eff}=\Phi_{int}\frac{\Phi_{f}+2\Phi_{int}-2\varphi_{s}\left(\Phi_{int}-\Phi_{f}\right)}{\Phi_{f}+2\Phi_{int}+\varphi_{s}\left(\Phi_{int}-\Phi_{f}\right)}$$
where *φ**_s_* is the filler volume fraction in the pseudo-dispersed phase, which in the case of 2D laminar particles (as in this study) can be estimated by the equation
(13)$$\varphi_{s}=\frac{\varphi_{f}}{\varphi_{f}\;+\;\varphi_{int}}=\;\frac{w}{w+2l_{int}}$$
where *φ**_int_* is the volume fraction of the filler in the interlayer region, which is a parameter that cannot be experimentally determined, and *w* and *l_int_* are the thicknesses of the filler and the interface, respectively. Note that the interface layer is formed on both sides of the filler sheets.

With this information, the effective permeability of the nanocomposites can be estimated by using a corrected Maxwell equation
(14)$$\Phi_{NC}=\Phi_{p}\frac{\Phi_{eff}+2\Phi_{p}-2\left(\varphi_{f}+\varphi_{int}\right)\left(\Phi_{p}-\Phi_{eff}\right)}{\Phi_{eff}+2\Phi_{p}+\left(\varphi_{f}+\varphi_{int}\right)\left(\Phi_{p}-\Phi_{eff}\right)}$$

With the composition of the nanocomposites studied and using a filler thickness of 10 nm and interface thicknesses from 0 to 100 nm, we can simulate the variation of the permeability of the nanocomposites for different values of β, and the results are presented in Figure 10**.** For this simulation we focused on the materials with 1 wt % of filler. As the fillers differ (G and GPE), the graphene volume fraction in the nanocomposites are also different due to the contribution of the pendant short PE brushes in GPE. Figure 10 represents the variation of permeability to oxygen of the nanocomposites with the size of the rigidized region at different β values, for the nanocomposites with G (Figure 10A) and GPE (Figure 10B).

With the linear fits from the points in Figure 10, and the experimental values of permeability of G1S, GPE1S, and GPE1M, we can assign a specific value of β and *l_int_* to adjust to the curves that provides an estimation of the extent of the rigidized region around the filler. In the evaluated range of β (2–50) the *l_int_* adopts values in the ranges of 3.4–8 nm, 17.5–39 nm, and 36.5–80 nm for G1S, GPE1S, and GPE1M, respectively. The variation of *l_int_* with β for these samples is represented in Figure 11.

From Figure 11, it can clearly be seen that the extent of the rigidized polymer region (*l_int_*) around the filler is greater for the nanocomposites with GPE, which is a clear indication of better compatibility and stronger polymer/filler interface when graphene is appropriately functionalized. In addition, a greater effect is observed for the samples prepared by melt-compounding which may be explained in terms of some preferential orientation of the filler in the direction of extrusion. In the case of samples prepared in solution, the filler is more homogeneously dispersed in a random manner throughout the polymer matrix.

Although the results for oxygen permeability are shown, the same trend has also been obtained for nitrogen and carbon dioxide (not shown).

Mathematical models have been proposed to describe the influence of the geometry and orientation of impermeable fillers on the barrier properties of different nanocomposites materials [47,48,49,50]. Such approaches describe the fillers as 2D or 3D plate-like shape (ribbons, flakes or disks), non-oriented (random distribution) or oriented in a perpendicular/parallel direction with respect to the diffusion path.

The dispersed impermeable nanofillers provoke a reduction in the diffusion coefficient of the composites (*D_NC_*), as shown in Figure 7B, since the gasses have to follow a more tortuous pathway through the membrane, in such a way that
(15)$$D_{NC}=\frac{D_{P}}{\tau }$$
where *D_P_* is the diffusion coefficient of the pristine polymer and *τ* is the tortuosity factor. If we assume that the presence of the filler does not affect the permeation properties of the polymer matrix, the gas solubility can be expressed as
(16)$$S_{NC}=S_{P}\;\left(1-\varphi_{f}\right)$$

By combining Equations (9), (15), and (16), the relative permeability of the nanocomposite in relation to the neat polymer is obtained by
(17)$$\frac{\Phi_{NC}}{\Phi_{P}}=\;\frac{1-\varphi_{f}}{\tau }$$

Nielsen suggested a model based on the dispersion of impermeable ribbons of infinite length in a permeable matrix [51]. This approximation describes the increase in the tortuosity of gas diffusion with the aspect ratio of the graphene layer (*α*) and the volume fraction of filler, according to
(18)$$\tau =1+\frac{\alpha \varphi_{f}}{2}$$

Thus, Equation (17) can be expressed as
(19)$$\frac{\Phi_{NC}}{\Phi_{P}}=\left(1-\varphi_{f}\right)/\left(1+\frac{\alpha \varphi_{f}}{2}\right)$$

This theoretical model assumes a complete exfoliation of the nanoplatelets dispersed along the perpendicular direction of gas diffusion, and can predict the experimental results accurately when *φ_f_* < 0.01 whereas for higher volume fraction of graphene layers, the filler tends to aggregate and this model is not valid. In this last case, Cussler et al. [52,53] proposed a model for ribbons of graphene oriented perpendicular to the direction of penetrant diffusion or in a random orientation where the tortuosity factor can be expressed as
(20)$$\tau =1+\frac{2\alpha \varphi_{f}}{3}+\frac{{\left(\alpha \varphi_{f}\right)}^{2}}{9}$$

As an example, Figure 12A shows the effect of the aspect ratio (*α*) on the permeability decrease as a function of the filler volume fraction for Nielsen and Cussler models. The experimental values for nanocomposites prepared with GPE filler are also represented in Figure 12. It is clear that for both models, the barrier performance improves as the value of *α* increases. Furthermore, our experimental data showed good agreements with the Nielsen model with an aspect ratio *α* = 10–20 and with the Cussler model with *α* = 10.

However, none of these models take into account the orientation of the platelets into the film thickness. Bharadwaj modified the Nielsen model including an orientation factor, *S*, which allows the calculation of zero and maximum tortuosity states [54]
(21)$$\frac{\Phi_{NC}}{\Phi_{P}}=\left(1-\varphi_{f}\right)/\left[1+\left(\frac{\alpha \varphi_{f}}{2}\right)\left(\frac{2}{3}\right)\left(S+\frac{1}{2}\right)\;\right]$$

This orientation factor takes values from −0.5 to 1. A value of *S* = −0.5 corresponds to the zero tortuosity case with the platelets oriented in a parallel direction respect to the gas diffusion. A value of *S* = 1 represents the maximum tortuosity with a perfect alignment of the platelets in the perpendicular orientation to the permeation direction. A value of *S* = 0 corresponds to a random distribution of graphene layers.

In the same way, we can extend the Cussler model by including this orientation parameter
(22)$$\frac{\Phi_{NC}}{\Phi_{P}}=\left(1-\varphi_{f}\right)/\left[1+\left(\frac{2\alpha \varphi_{f}}{3}+\frac{{\left(\alpha \varphi_{f}\right)}^{2}}{9}\right)\left(\frac{2}{3}\right)\;\left(S+\frac{1}{2}\right)\right]$$

Taking into account that the volume fractions of GPE used to prepare our nanocomposite membranes were higher than 0.01, which is the upper limit for achieving better applicability of the Nielsen model, we have compared our experimental data with the modified Cussler model of Equation (22), using *α* = 10 and different values for *S*. The results are shown in Figure 12B. As can be seen, the decrease in the permeability of GPE1S membrane appears to satisfy Cussler model with a value of *S* = 0, whereas GPE1M and GPE3M fit very well with this model with 0.5 < *S* < 1. These results would mean that for membranes prepared from solution, the functionalized graphene platelets have a random orientation with respect to the permeation direction, while when melt-blending was employed, the graphene sheets seems to be partially aligned with the film surface. These results are in good agreement with the TEM images of Figure 2.

Summarizing, the much lower permeability of the nanocomposites with GPE can be unequivocally attributed to the better compatibility of GPE with the matrix caused by strong interactions between the pendant short PE brushes in graphene with the HDPE, and the differences between samples prepared by different experimental approaches can be attributed to different dispersion, interphase morphology, and some orientation of the fillers during the sample preparation process.

### 3.5. Electrical Properties

One of the main challenges for systems composed of polyolefins and highly conductive particles like graphene is to achieve polymer nanocomposites with reasonable electrical conductivity. Polyolefins are totally insulating, with conductivity values in the order of 10^−14^ S cm^−1^. Therefore, it is expected that the incorporation of small amounts of graphene with high aspect ratio can produce a significant increase in the electrical conductivity of the host polymer matrices. In fact, for graphene-based polymer nanocomposites with different polymer matrices, it has been shown that the electrical conductivity of the insulating polymer can be increased by several orders of magnitude in a percolative manner with the addition of critical amounts of graphene. The critical loading content, at which the conductivity suddenly increases, known as the electrical percolation threshold, depends on the type of matrix, the characteristics of graphene (structural quality, surface functionalization, lateral dimensions, thickness, etc.), the polymer/filler affinity and the mixing procedure. For certain matrices, the use of reduced graphene oxide has demonstrated to be very useful to achieve electrical conductivity [5,55]. For polyolefins this approach is, in principle, not the preferred one as no strong polymer filler interactions are expected. However, solution mixing with either unmodified or chemically functionalized graphene oxide followed by reduction has been shown to produce polyethylene nanocomposites with electrical conductivity [22,25].

In Table 5, we present the electrical conductivity values, obtained by the four-probe method, for all samples prepared in this study. From Table 5, we can clearly see the influence of the type of the filler and mixing methods. Firstly, we can remark that in this class of nanocomposites, the use of unmodified graphene as filler does not produce materials with good electrical conductivity. In fact, in previous studies it was shown that in nanocomposites of HDPE with higher amounts of unmodified graphene no electrical conductivity was obtained [20]. This can be attributed to a poor dispersion of graphene and/or to weak or null filler/polymer interfacial interactions in this system. This is supported by results in Table 5, where measurable conductivity values for nanocomposites with GPE are presented. Among this family of nanocomposites it can clearly be seen that the nanocomposites prepared by solution mixing present higher values than those prepared by melt-compounding. This can be attributed not only to the better polymer/filler interface due to a more intimate contact between both components, but also to a better dispersion of the filler. In samples prepared by solution mixing and precipitation, the filler is randomly oriented within the nanocomposites, which promotes the formation of a three-dimensional conductive network that facilitates the percolation phenomenon with low amounts of filler. For melt-compounded samples, as the amount of graphene increases some agglomerates are observed and at the highest content some orientation of the filler along the extrusion direction seems to develop, producing materials with certain anisotropy. However, this preferential orientation reduces the contact between fillers in the rest of directions, significantly increasing the critical loading content at which electrical percolation occurs. Normally for melt compounding via extrusion and injection molding of HDPE/GNP nanocomposites, the percolation threshold typically lies in the range of 10–15% [56], but selective aggregation during processing has been shown to lead to conductive pathways that can reduce this threshold to between 3–5%. Segregated structures can be very effective to produce a conductive network and a very low percolation threshold (0.07% GNP) and high conductivity (10^−2^ S cm^−1^ with 0.6% GNP) has been reported for ultra-high molecular weight polyethylene-GNP nanocomposites produced via solvent assisted dispersion and melt compression [57].

At this point, it is worth noting that while the strength of the filler/polymer interface is fundamental to achieve nanocomposites with improved or new properties, the mixing approaches can produce opposite effects in electrical conductivity and gas barrier. While in the former the orientation of the filler can be detrimental for the electron transport through the nanocomposite, in the latter this is helpful to increase the tortuous path limiting the diffusion of gas molecules.

## 4. Conclusions

The preparation of multifunctional nanocomposites by combination of HDPE and graphene has been described. The adequate combination of both components produces nanocomposites with better thermal and mechanical properties incorporating gas barrier and electrical conductivity. The importance of an appropriate selection of the type of filler and the mixing method for the preparation of HDPE/graphene nanocomposites has been highlighted. Consistent results demonstrate a clear influence of both parameters on the final properties of the nanocomposites. While thermal and mechanical properties are slightly affected, gas barrier and electrical conductivity are strongly dependent on the type of filler as well as on the mixing approach. While melt-compounding produces better membranes for gas barrier due to some orientation along the extrusion direction, this is disadvantageous for electrical conductivity at the graphene contents tested. Furthermore, the mechanical and barrier properties of the materials prepared here make them good candidates for food packaging, although in this particular case, issues related to reduced transparency must be addressed.

## Figures and Tables

**Figure 1 polymers-12-02094-f001:**
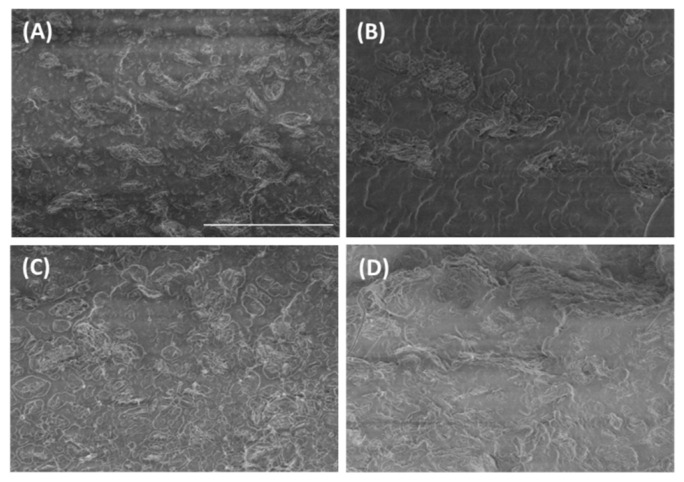
SEM images of (**A**) G1M, (**B**) G1S, (**C**) GPE1M, (**D**) GPE3M. The scale bar applies for all samples and corresponds to 10 µm.

**Figure 2 polymers-12-02094-f002:**
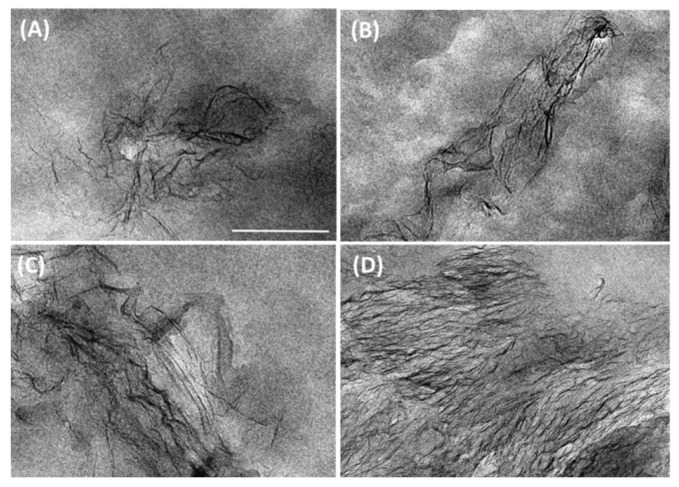
TEM images of microtomed thin sections of (**A**) G1M, (**B**) G1S, (**C**) GPE1M, (**D**) GPE3M. The scale bar applies for all samples and corresponds to 200 nm.

**Figure 3 polymers-12-02094-f003:**
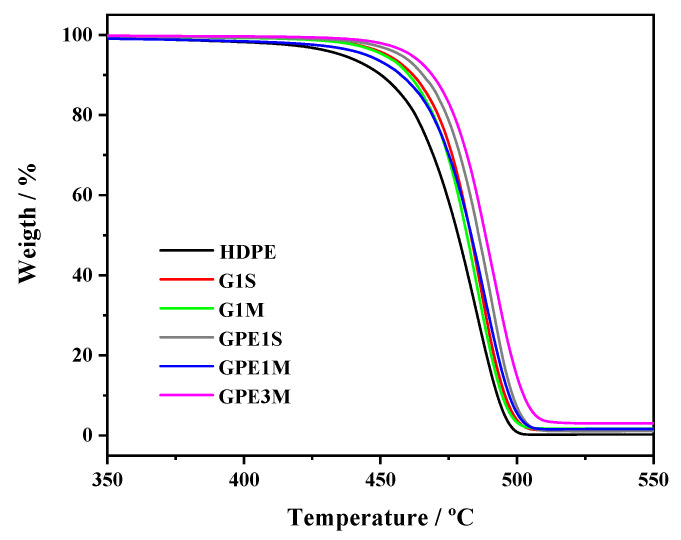
TG curves under a nitrogen atmosphere at a rate of 10 °C min^−1^ for the different samples.

**Figure 4 polymers-12-02094-f004:**
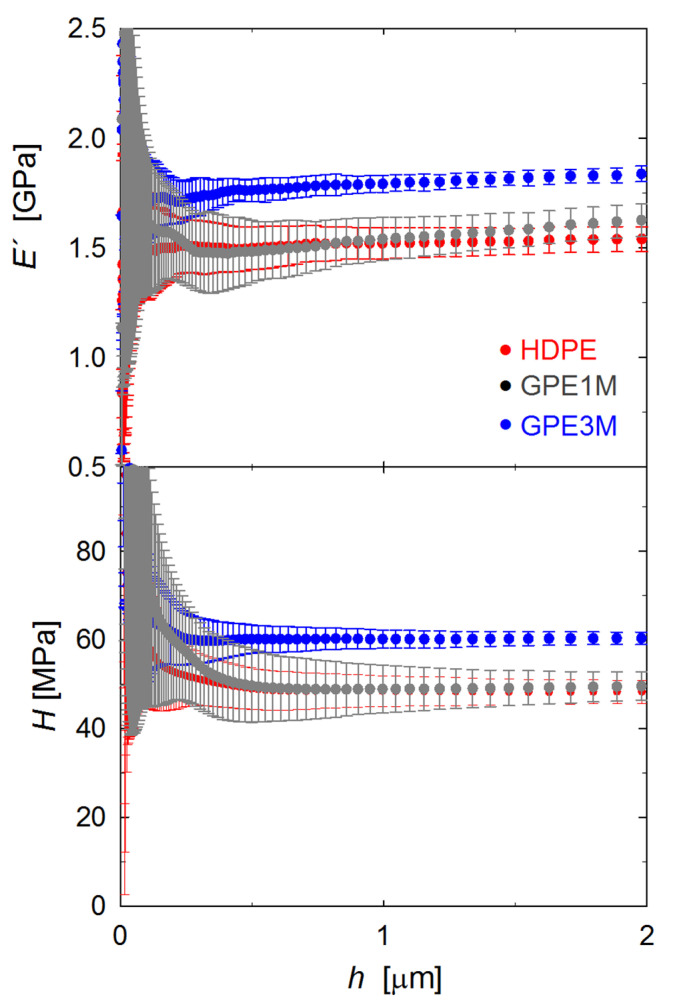
Storage modulus, *E߰*, and hardness, *H*, as a function of indenter displacement for HDPE and the two nanocomposites with modified graphene prepared using melt extrusion.

**Figure 5 polymers-12-02094-f005:**
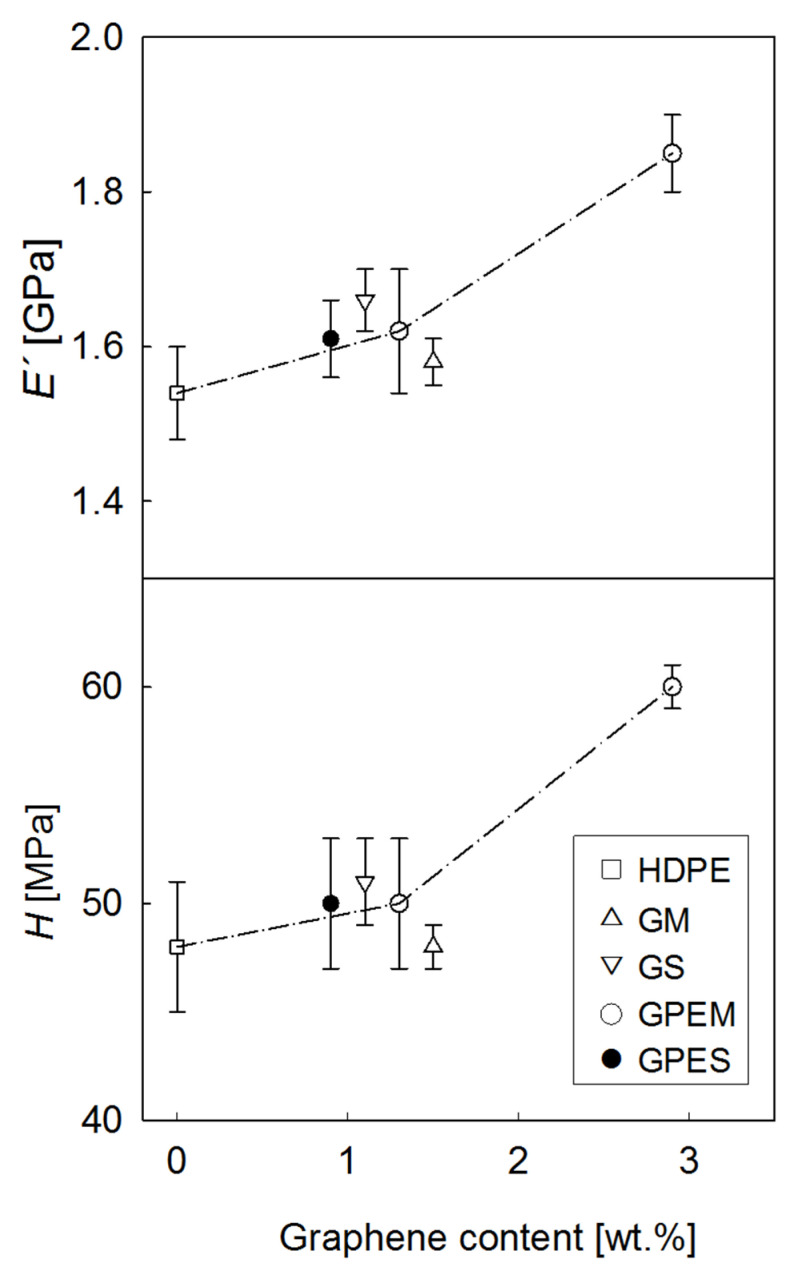
Storage modulus, *E′* and hardness, *H*, as a function of graphene content (neat graphene, i.e., excluding the polymer brushes) for all the nanocomposites and the starting HDPE: GM, neat graphene and melt extruded; GS, neat graphene and solution mixed; GPEM, modified graphene and melt extruded; GPES, modified graphene and solution mixed. The dashed lines are guides for the eyes.

**Figure 6 polymers-12-02094-f006:**
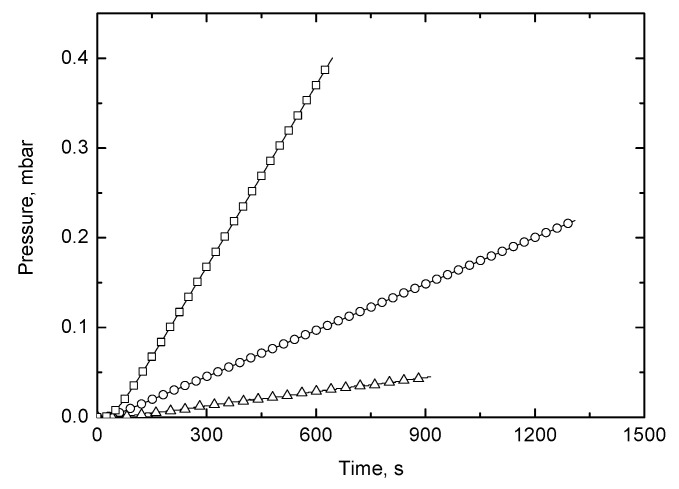
Evolution of the pressure of oxygen (**○**), nitrogen (**△**), and carbon dioxide (□) in the downstream chamber as a function of time for the GPE1M membrane. Symbols and continuous lines correspond to the experimental and calculated values from Equation (5), respectively.

**Figure 7 polymers-12-02094-f007:**
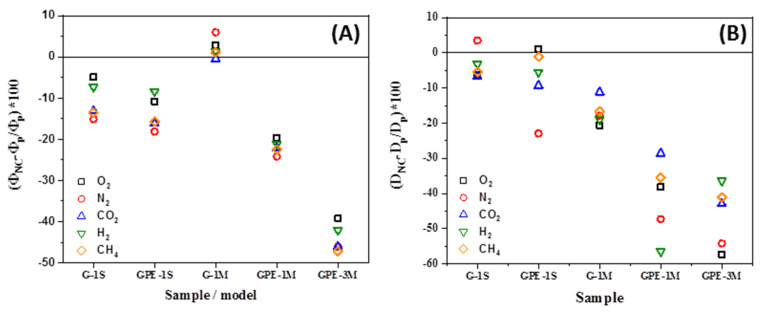
Improvement of the gas barrier properties (**A**) and variation of the diffusion coefficient (**B**) of the nanocomposites with respect to a HDPE membrane.

**Figure 8 polymers-12-02094-f008:**
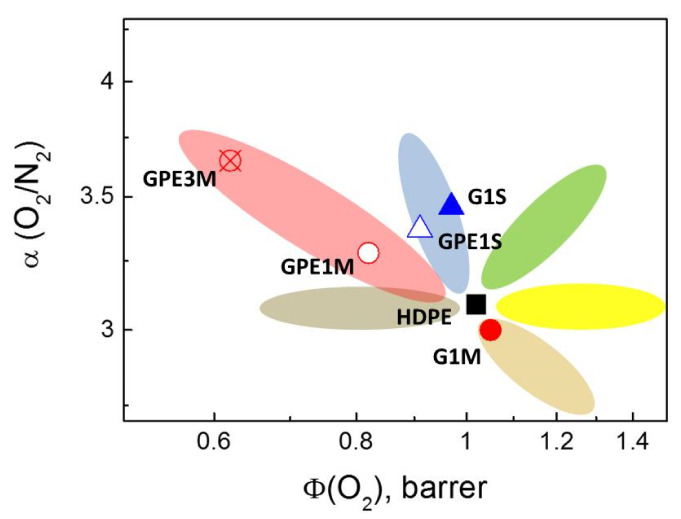
Representation of the relationship between composite membranes morphologies and gas transport. Symbols represent experimental data for the membranes studied in this work: (■) HDPE, (●) G1M, (▲) G1S, (△) GPE1S, (○) GPE1M, and (⊗) GPE3M.

**Figure 9 polymers-12-02094-f009:**
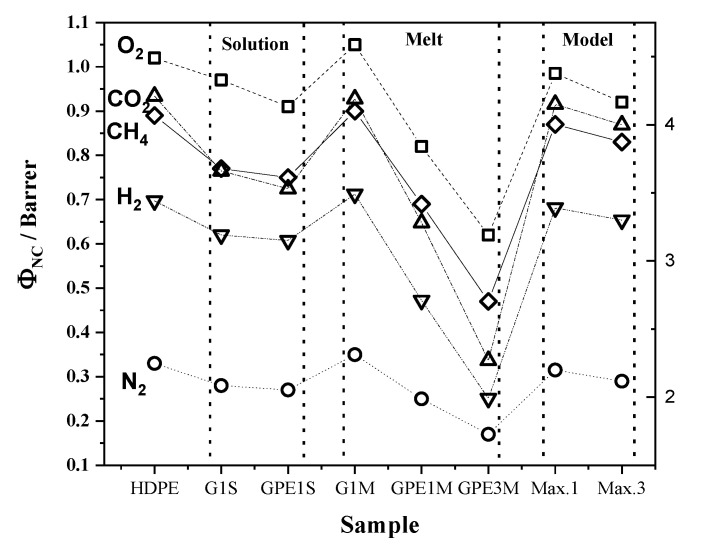
Comparison of the experimental permeability for all samples tested with the theoretical prediction using the Maxwell model.

**Figure 10 polymers-12-02094-f010:**
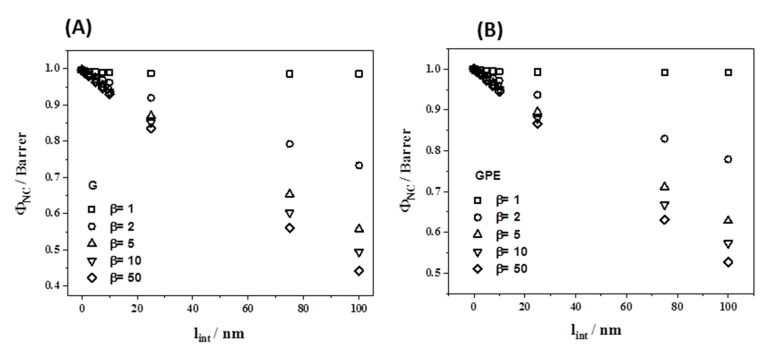
Dependence of Φ*_NC_* for oxygen permeation with the interface thickness (*l_int_*) of rigidified polymer regions around the filler for the nanocomposites with G (**A**) and GPE (**B**). The simulation was made using the experimental value of 1.02 Barrer for Φ*_p_*.

**Figure 11 polymers-12-02094-f011:**
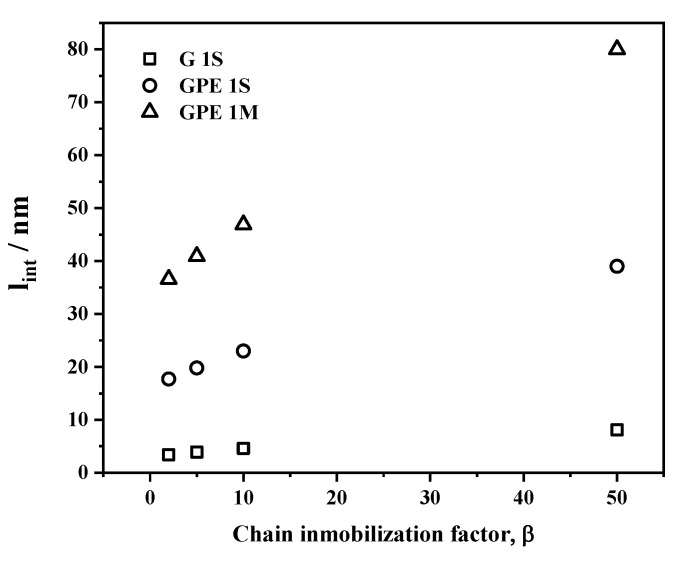
Plot comparing the extent of the interface for representative samples, and its dependence on β.

**Figure 12 polymers-12-02094-f012:**
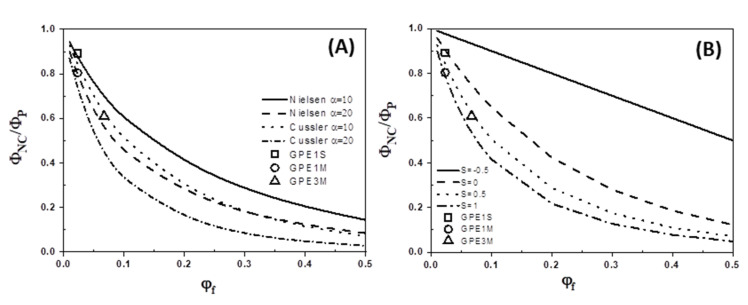
(**A**) Effect of aspect ratio of graphene layers on the barrier properties of nanocomposites as a function of the filler volume fraction for Nielsen and Cussler models. Experimental data of GPE1S, GPE1M, and GPE3M are also represented, using Φ*_P_* = 1.02 Barrer. (**B**) Comparison between experimental permeability for GPE1S, GPE1M, and GPE3M with the model predictions from Equation (22), with Φ*_P_* = 1.02 Barrer, *α* = 10 for different values of tortuosity parameter, *S*.

**Table 1 polymers-12-02094-t001:** Thermal parameters obtained from TGA and DSC.

Sample	G Content (wt %) ^a^	*T_i_* (°C)	*T*_10_ (°C)	*T_max_* (°C)	*X_c_* (%)	*T_m_* (°C)
HDPE	-	406 ± 1	450 ± 1	485 ± 1	58 ± 2	129.3 ± 0.5
G1M	1.5	436 ± 1	460 ± 1	485 ± 1	55 ± 2	129. 4 ± 0.5
GPE1M	1.3	415 ± 1	457 ± 1	488 ± 1	60 ± 2	130.0 ± 0.5
GPE3M	2.9	450 ± 1	469 ± 1	491 ± 1	58 ± 2	127.3 ± 0.5
G1S	1.1	437 ± 1	462 ± 1	487 ± 1	57 ± 2	128.5 ± 0.5
GPE1S	0.9	444 ± 1	466 ± 1	490 ± 1	59 ± 2	129.6 ± 0.5

^a^ Graphene content measured by TGA after heating to 800 °C. *T_i_* = initial degradation temperature obtained at 2% weight loss, *T*_10_ = temperature corresponding to 10% weight loss, *T_max_* = maximum degradation rate temperature. *X_c_* and *T_m_* are the degree of crystallinity and melting temperature measured by DSC.

**Table 2 polymers-12-02094-t002:** *E′* and *H* derived from instrumented indentation at a penetration depth of 2 μm.

Sample	E′ (GPa)	H (MPa)
HDPE	1.54 ± 0.06	48 ± 3
G1M	1.58 ± 0.03	48 ± 1
GPE1M	1.62 ± 0.08	50 ± 3
G1S	1.66 ± 0.04	51 ± 2
GPE1S	1.61 ± 0.05	50 ± 3
GPE3M	1.85 ± 0.05	60 ± 1

**Table 3 polymers-12-02094-t003:** Values of the permeability (Barrer), diffusion (cm^2^ s^−1^) and apparent solubility (cm^3^/cm^3^ cmHg) coefficients of different gases at 30 °C and 1 bar of pressure in HDPE and nanocomposite membranes used in this study and prepared from solution.

Gas	HDPE			G1S			GPE1S		
	Φ	D × 10^7^	S × 10^4^	Φ	D × 10^7^	S × 10^4^	Φ	D × 10^7^	S × 10^4^
O_2_	1.02	2.72	3.76	0.97	2.55	3.79	0.91	2.75	3.32
N_2_	0.33	2.01	1.62	0.28	2.08	1.35	0.27	1.55	1.72
CO_2_	4.21	1.61	26.08	3.66	1.50	24.44	3.53	1.46	24.24
H_2_	3.44	25.41	1.35	3.19	24.59	1.30	3.15	24.0	1.31
CH_4_	0.89	0.90	9.92	0.77	0.85	8.97	0.75	0.89	8.38

**Table 4 polymers-12-02094-t004:** Values of the permeability (Barrer), diffusion (cm^2^ s^−1^), and apparent solubility (cm^3^/cm^3^ cmHg) coefficients of different gases at 30 °C and 1 bar of pressure in the nanocomposite membranes used in this study and prepared by melt-blending.

Gas	G1M			GPE1M			GPE3M		
	Φ	D × 10^7^	S × 10^4^	Φ	D × 10^7^	S × 10^4^	Φ	D × 10^7^	S × 10^4^
O_2_	1.05	2.16	4.85	0.82	1.68	4.88	0.62	1.16	5.34
N_2_	0.35	1.65	2.11	0.25	1.06	2.37	0.17	0.92	1.86
CO_2_	4.19	1.43	29.39	3.28	1.15	28.37	2.27	0.92	24.78
H_2_	3.49	20.57	1.70	2.71	11.05	2.45	1.99	16.16	1.23
CH_4_	0.90	0.75	12.02	0.69	0.58	11.74	0.47	0.53	8.90

**Table 5 polymers-12-02094-t005:** Electrical conductivity measured by four-probe technique, for all samples prepared in this study.

Sample	Electrical Conductivity (S cm^−1^)
G1S	<1 × 10^−8^ *
GPE1S	~1 × 10^−7^
G1M	<1 × 10^−8^ *
GPE1M	<1 × 10^−8^ *
GPE3M	4.9 × 10^−6^

* 1 × 10^−8^ is the limit of detection of the set-up used to measure the conductivity values.

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
