# Peer review of "Graphene and Polyethylene: A Strong Combination Towards Multifunctional Nanocomposites"

_polymers, 2020, doi:10.3390/polym12092094_

Round 1
Reviewer 1 Report
In this article, authors developed a series of nanocomposites of graphene and a polyolefin using different experimental approaches. I suggest to publish the research after minor revision.
1) where is the scale bar for Fig.2?
2) How many samples have been measured in TGA and DSC tests? Where is the error bar for the values in Table 1?
Author Response
In this article, authors developed a series of nanocomposites of graphene and a polyolefin using different experimental approaches. I suggest to publish the research after minor revision.
- where is the scale bar for Fig.2?
Authors’ reply. The scale bar has been added to Figure 2.
- How many samples have been measured in TGA and DSC tests? Where is the error bar for the values in Table 1?
Authors’ reply. We thank the reviewer for this comment. Error bars for all parameters have been added to the values in Table 1
Reviewer 2 Report
Article entitled "Graphene and polyethylene: a strong combination towards multifunctional nanocomposites" by Mar López-González, Araceli Flores, Fabrizio Marra, Gary Ellis, Marián Gómez, Horacio Salavagione concerns the preparation of a series of nanocomposites of graphene and a polyolefin using different experimental approaches, with the final goal of obtaining multifunctional materials. A high-density polyethylene (HDPE) as the matrix was used in the study, while unmodified and chemically modified graphene fillers are employed. In my opinion, these are interesting issues concerning the preparation of the homogeneous dispersion of the filler. In general, the article is written correctly, the technical resources used for the research have been presented, the methodology is clearly indicated, the structure of the article is correct, although the authors have not avoided certain inaccuracies and minor shortcomings that need to be corrected or supplemented in order to publish the article. I include these defects:
- - p. 3, the MFI should be given as the mass flow rate (MFR), and the temperature of the determination should be given in addition to the load,
- - p. 4, in the case of testing the dispersion of the fillers in the HDPE matrix, the description should be slightly expanded to include the type of detector and the acceleration voltage.
- - under point 3: Results and discussion. Points 3.1, 3.2 and 3.3 have no discussion. Please conduct such a discussion in each of these points and refer to/support the interpretation of the results received to the results received by other authors,
- - It seems that all the pictures are too big, and the font in the tables is also too big, - Please correct the description of the vertical axis in Figures 4 and 5 (it is E?),
- - please correct the numbering of the literature inventory.
After these defects have been corrected, the article can be published in the journal "Polymers". Best regards,
Author Response
Article entitled "Graphene and polyethylene: a strong combination towards multifunctional nanocomposites" by Mar López-González, Araceli Flores, Fabrizio Marra, Gary Ellis, Marián Gómez, Horacio Salavagione concerns the preparation of a series of nanocomposites of graphene and a polyolefin using different experimental approaches, with the final goal of obtaining multifunctional materials. A high-density polyethylene (HDPE) as the matrix was used in the study, while unmodified and chemically modified graphene fillers are employed. In my opinion, these are interesting issues concerning the preparation of the homogeneous dispersion of the filler. In general, the article is written correctly, the technical resources used for the research have been presented, the methodology is clearly indicated, the structure of the article is correct, although the authors have not avoided certain inaccuracies and minor shortcomings that need to be corrected or supplemented in order to publish the article. I include these defects:
- 3, the MFI should be given as the mass flow rate (MFR), and the temperature of the determination should be given in addition to the load,
Authors’ reply. P.3.; section 2.1; second paragraph. MFI has been replaced by mass flow rate (MFR), and the temperature of determination (190ºC) has been included.
- 4, in the case of testing the dispersion of the fillers in the HDPE matrix, the description should be slightly expanded to include the type of detector and the acceleration voltage.
Authors’ reply. P.4; first paragraph. The information on the voltage employed and the type of detection has been added.
- under point 3: Results and discussion. Points 3.1, 3.2 and 3.3 have no discussion. Please conduct such a discussion in each of these points and refer to/support the interpretation of the results received to the results received by other authors,
Authors’ reply. The results in sections 3.1, 3.2 and 3.3 have been compared to those reported for graphene/HDPE nanocomposites and the discussion has been expanded as follow:
Section 3.1; P.6. A paragraph relating our results with previous ones has been added at the end of section 3.1.
Section 3.2; P.7. New paragraphs comparing the thermal behavior of our materials with results in the literature has been included in page 7, before Table 1 and before Figure 3.
Section 3.3; P. 9 (before Table 2). The paragraph has been rewritten in order to include a comparison with previous works.
- It seems that all the pictures are too big, and the font in the tables is also too big, - Please correct the description of the vertical axis in Figures 4 and 5 (it is E?),
Authors’ reply. The size of Figure 1 and 2 has been reduced to adjust to the wide of a single column. The font size in all tables has been reduced to 10 ppt.
Regarding the y-axis in Figures 4 and 5, it is correct as it refers to the storage modulus (represented by E´) and not to the Young’s modulus (denoted by E). We have now clearly indicated in the captions of Figures 4 and 5 that E´ refers to the storage modulus
- please correct the numbering of the literature inventory.
Authors’ reply. The numbering has been corrected; a non-numbered reference, listed in the reference section (page 22, after Ref. 47) has been removed.